# Stochastic Proximal Gradient Descent with Acceleration Techniques

**Atsushi Nitanda**
NTT DATA Mathematical Systems Inc.
1F Shinanomachi Rengakan, 35,
Shinanomachi, Shinjuku-ku, Tokyo,
160-0016, Japan
nitanda@msi.co.jp

## Abstract

Proximal gradient descent (PGD) and stochastic proximal gradient descent (SPGD) are popular methods for solving regularized risk minimization problems in machine learning and statistics. In this paper, we propose and analyze an accelerated variant of these methods in the mini-batch setting. This method incorporates two acceleration techniques: one is Nesterov's acceleration method, and the other is a variance reduction for the stochastic gradient. Accelerated proximal gradient descent (APG) and proximal stochastic variance reduction gradient (Prox-SVRG) are in a trade-off relationship. We show that our method, with the appropriate mini-batch size, achieves lower overall complexity than both APG and Prox-SVRG.

## 1 Introduction

This paper consider the following optimization problem:

$$\underset{x \in \mathbb{R}^d}{\text{minimize}} \ f(x) \overset{\text{def}}{=} g(x) + h(x), \tag{1}$$

where $g$ is the average of the smooth convex functions $g_1, \ldots, g_n$ from $\mathbb{R}^d$ to $\mathbb{R}$, i.e., $g(x) = \frac{1}{n} \sum_{i=1}^{n} g_i(x)$ and $h : \mathbb{R}^d \to \mathbb{R}$ is a relatively simple convex function that can be non-differentiable. In machine learning, we often encounter optimization problems of this form. For example, given a sequence of training examples $(a_1, b_1), \ldots, (a_n, b_n)$, where $a_i \in \mathbb{R}^d$ and $b_i \in \mathbb{R}$, if we set $g_i(x) = \frac{1}{2}(a_i^T x - b_i)^2$, then we obtain ridge regression by setting $h(x) = \frac{\lambda}{2}\|x\|^2$ or we obtain Lasso by setting $h(x) = \lambda|x|$. If we set $g_i(x) = \log(1 + \exp(-b_i x^T a_i))$, then we obtain regularized logistic regression.

To solve the optimization problem (1), one popular method is proximal gradient descent (PGD), which can be described by the following update rule for $k = 1, 2, \ldots$:

$$x_{k+1} = \text{prox}_{\eta_k h}\left(x_k - \eta_k \nabla g(x_k)\right),$$

where prox is the proximity operator,

$$\text{prox}_{\eta h}(y) = \underset{x \in \mathbb{R}^d}{\arg\min}\left\{\frac{1}{2}\|x - y\|^2 + \eta h(x)\right\}.$$

A stochastic variant of PGD is stochastic proximal gradient descent (SPGD), where at each iteration $k = 1, 2, \ldots$, we pick $i_k$ randomly from $\{1, 2, \ldots, n\}$, and take the following update:

$$x_{k+1} = \text{prox}_{\eta_k h}\left(x_k - \eta_k \nabla g_{i_k}(x_k)\right).$$

The advantage of SPGD over PGD is that at each iteration, SPGD only requires the computation of a single gradient $\nabla g_{i_k}(x_k)$. In contrast, each iteration of PGD evaluates the $n$ gradients. Thus the computational cost of SPGD per iteration is $1/n$ that of the PGD. However, due to the variance introduced by random sampling, SPGD obtains a slower convergence rate than PGD. In this paper we consider problem (1) under the following assumptions.

**Assumption 1.** *Each convex function $g_i(x)$ is $L$-Lipschitz smooth, i.e., there exist $L > 0$ such that for all $x, y \in \mathbb{R}^d$,*

$$\|\nabla g_i(x) - \nabla g_i(y)\| \leq L\|x - y\|. \tag{2}$$

*From (2), one can derive the following inequality,*

$$g_i(x) \leq g_i(y) + \langle \nabla g_i(y), x - y \rangle + \frac{L}{2}\|x - y\|^2. \tag{3}$$

**Assumption 2.** *$g(x)$ is $\mu$-strongly convex; i.e., there exists $\mu > 0$ such that for all $x, y \in \mathbb{R}^d$,*

$$g(x) \geq g(y) + \langle \nabla g(y), x - y \rangle + \frac{\mu}{2}\|x - y\|^2. \tag{4}$$

*Note that it is obvious that $L \geq \mu$.*

**Assumption 3.** *The regularization function $h(x)$ is a lower semi-continuous proper convex function; however, it can be non-differentiable or non-continuous.*

Under the Assumptions 1, 2, and $h(x) \equiv 0$, PGD (which is equivalent to gradient descent in this case) with a constant learning rate $\eta_k = \frac{1}{L}$ achieves a linear convergence rate. On the other hand, for stochastic (proximal) gradient descent, because of the variance introduced by random sampling, we need to choose diminishing learning rate $\eta_k = O(1/k)$, and thus the stochastic (proximal) gradient descent converges at a sub-linear rate.

To improve the stochastic (proximal) gradient descent, we need a variance reduction technique, which allows us to take a larger learning rate. Recently, several papers proposed such variance reduction methods for the various special cases of (1). In the case where $g_i(x)$ is Lipschitz smooth and $h(x)$ is strongly convex, Shalev-Shwartz and Zhang [1, 2] proposed a proximal stochastic dual coordinate ascent (Prox-SDCA); the same authors developed accelerated variants of SDCA [3, 4]. Le Roux et al. [5] proposed a stochastic average gradient (SAG) for the case where $g_i(x)$ is Lipschitz smooth, $g(x)$ is strongly convex, and $h(x) \equiv 0$. These methods achieve a linear convergence rate. However, SDCA and SAG need to store all gradients (or dual variables), so that $O(nd)$ storage is required in general problems. Although this can be reduced to $O(n)$ for linear prediction problems, these methods may be unsuitable for more complex and large-scale problems. More recently, Johnson and Zhang [6] proposed stochastic variance reduction gradients (SVRG) for the case where $g_i(x)$ is $L$-Lipschitz smooth, $g(x)$ is $\mu$-strongly convex, and $h(x) \equiv 0$. SVRG achieves the following overall complexity (total number of component gradient evaluations to find an $\epsilon$-accurate solution),

$$O\left((n + \kappa)\log\left(\frac{1}{\epsilon}\right)\right), \tag{5}$$

where $\kappa$ is the condition number $L/\mu$. Furthermore, this method need not store all gradients. Xiao and Zhang [7] proposed a proximal variant of SVRG, called Prox-SVRG which also achieves the same complexity.

Another effective method for solving (1) is accelerated proximal gradient descent (APG), proposed by Nesterov [8, 9]. APG [8] is an accelerated variant of deterministic gradient descent and achieves the following overall complexity to find an $\epsilon$-accurate solution,

$$O\left(n\sqrt{\kappa}\log\left(\frac{1}{\epsilon}\right)\right). \tag{6}$$

Complexities (5) and (6) are in a trade-off relationship. For example, if $\kappa = n$, then the complexity (5) is less than (6). On the other hand, the complexity of APG has a better dependence on the condition number $\kappa$.

In this paper, we propose and analyze a new method called the Accelerated Mini-Batch Prox-SVRG (Acc-Prox-SVRG) for solving (1). Acc-Prox-SVRG incorporates two acceleration techniques in the mini-batch setting: (1) Nesterov's acceleration method of APG and (2) an variance reduction technique of SVRG. We show that the overall complexity of this method, with an appropriate mini-batch size, is more efficient than both Prox-SVRG and APG; even when mini-batch size is not appropriate, our method is still comparable to APG or Prox-SVRG.

## 2  Accelerated Mini-Batch Prox-SVRG

As mentioned above, to ensure convergence of SPGD, the learning rate $\eta_k$ has to decay to zero for reducing the variance effect of the stochastic gradient. This slows down the convergence. As a remedy to this issue, we use the variance reduction technique of SVRG [6] (see also [7]), which allows us to take a larger learning rate. Acc-Prox-SVRG is a multi-stage scheme. During each stage, this method performs $m$ APG-like iterations and employs the following direction with mini-batch instead of gradient,

$$v_k = \nabla g_{I_k}(y_k) - \nabla g_{I_k}(\tilde{x}) + \nabla g(\tilde{x}), \tag{7}$$

where $I_k = \{i_1, \ldots, i_b\}$ is a randomly chosen size $b$ subset of $\{1, 2, \ldots, n\}$ and $g_{I_k} = \frac{1}{b}\sum_{j=1}^{b} g_{i_j}$. At the beginning of each stage, the initial point $x_1$ is set to be $\tilde{x}$, and at the end of stage, $\tilde{x}$ is updated. Conditioned on $y_k$, we can take expectation with respect to $I_k$ and obtain $\mathbb{E}_{I_k}[v_k] = \nabla g(y_k)$, so that $v_k$ is an unbiased estimator. As described in the next section, the conditional variance $\mathbb{E}_{I_k}\|v_k - \nabla g(y_k)\|^2$ can be much smaller than $\mathbb{E}_i\|\nabla g_i(y_k) - \nabla g(y_k)\|^2$ near the optimal solution. The pseudo-code of our Acc-Prox-SVRG is given in Figure 1.

---

**Parameters** update frequency $m$, learning rate $\eta$, mini-batch size $b$
 and non-negative sequence $\beta_1, \ldots, \beta_m$
**Initialize** $\tilde{x}_1$
**Iterate:**  for $s = 1, 2, \ldots$
 $\tilde{x} = \tilde{x}_s$
 $\tilde{v} = \frac{1}{n}\sum_{i=1}^{n} \nabla g_i(\tilde{x})$
 $x_1 = y_1 = \tilde{x}$
 **Iterate:**  for $k = 1, 2, \ldots, m$
   Randomly pick subset $I_k \subset \{1, 2, \ldots, n\}$ of size $b$
   $v_k = \nabla g_{I_k}(y_k) - \nabla g_{I_k}(\tilde{x}) + \tilde{v}$
   $x_{k+1} = \mathrm{prox}_{\eta h}\left(y_k - \eta v_k\right)$
   $y_{k+1} = x_{k+1} + \beta_k(x_{k+1} - x_k)$
 set $\tilde{x}_{s+1} = x_{m+1}$
 **end**
**end**

---

Figure 1: Acc-Prox-SVRG

In our analysis, we focus on a basic variant of the algorithm (Figure 1) with $\beta_k = \frac{1-\sqrt{\mu\eta}}{1+\sqrt{\mu\eta}}$.

## 3  Analysis

In this section, we present our analysis of the convergence rates of Acc-Prox-SVRG described in Figure 1 under Assumptions 1, 2 and 3, and provide some notations and definitions. Note that we may omit the outer index $s$ for notational simplicity. By the definition of a proximity operator, there exists a subgradient $\xi_k \in \partial h(x_{k+1})$ such that

$$x_{k+1} = y_k - \eta\left(v_k + \xi_k\right).$$

We define the *estimate sequence* $\Phi_k(x)$ $(k = 1, 2, \ldots, m+1)$ by

$$\Phi_1(x) = f(x_1) + \frac{\mu}{2}\|x - x_1\|^2 \ \ and$$

$$\Phi_{k+1}(x) = (1 - \sqrt{\mu\eta})\Phi_k(x) + \sqrt{\mu\eta}(g_{I_k}(y_k) + (v_k, x - y_k) + \frac{\mu}{2}\|x - y_k\|^2$$
$$+ h(x_{k+1}) + (\xi_k, x - x_{k+1})), \ \ for \ k \geq 1.$$

We set

$$\Phi_k^* = \min_{x \in \mathbb{R}^d} \Phi_k(x) \ \ and \ \ z_k = \arg\min_{x \in \mathbb{R}^d} \Phi_k(x).$$

Since $\nabla^2 \Phi_k(x) = \mu I_n$, it follows that for $\forall x \in \mathbb{R}^d$,

$$\Phi_k(x) = \frac{\mu}{2}\|x - z_k\|^2 + \Phi_k^*. \tag{8}$$

The following lemma is the key to the analysis of our method.

**Lemma 1.** *Consider Acc-Prox-SVRG in Figure 1 under Assumptions 1, 2, and 3. If $\eta \leq \frac{1}{2L}$, then for $k \geq 1$ we have*

$$\mathbb{E}\left[\Phi_k(x)\right] \leq f(x) + (1 - \sqrt{\mu\eta})^{k-1}(\Phi_1 - f)(x) \quad and \tag{9}$$

$$\mathbb{E}\left[f(x_k)\right] \leq \mathbb{E}\left[\Phi_k^* + \sum_{l=1}^{k-1}(1 - \sqrt{\mu\eta})^{k-1-l}\left\{-\frac{\mu}{2}\frac{1 - \mu\eta}{\sqrt{\mu\eta}}\|x_l - y_l\|^2 + \eta\|\nabla g(y_l) - v_l\|^2\right\}\right], \tag{10}$$

*where the expectation is taken with respect to the history of random variables $I_1, \ldots, I_{k-1}$.*

Note that if the conditional variance of $v_l$ is equal to zero, we immediately obtain a linear convergence rate from (9) and (10). Before we can prove Lemma 1, additional lemmas are required, whose proofs may be found in the Supplementary Material.

**Lemma 2.** *If $\eta < \frac{1}{\mu}$, then for $k \geq 1$ we have*

$$z_{k+1} = (1 - \sqrt{\mu\eta})z_k + \sqrt{\mu\eta}y_k - \sqrt{\frac{\eta}{\mu}}(v_k + \xi_k) \quad and \tag{11}$$

$$z_k - y_k = \frac{1}{\sqrt{\mu\eta}}(y_k - x_k). \tag{12}$$

**Lemma 3.** *For $k \geq 1$, we have*

$$(\nabla g(y_k) + \xi_k, v_k + \xi_k) = \frac{1}{2}\left(\|\nabla g(y_k) + \xi_k\|^2 + \|v_k + \xi_k\|^2 - \|\nabla g(y_k) - v_k\|^2\right), \tag{13}$$

$$\|v_k + \xi_k\|^2 \leq 2\left(\|\nabla g(y_k) + \xi_k\|^2 + \|\nabla g(y_k) - v_k\|^2\right), \quad and \tag{14}$$

$$\|\nabla g(y_k) + \xi_k\|^2 \leq 2\left(\|v_k + \xi_k\|^2 + \|\nabla g(y_k) - v_k\|^2\right). \tag{15}$$

*Proof of Lemma 1.* Using induction, it is easy to show (9). The proof is in Supplementary Material. Now we prove (10) by induction. From the definition of $\Phi_1$, $\Phi_1^* = f(x_1)$. we assume (10) is true for $k$. Using Eq. (11), we have

$$\|y_k - z_{k+1}\|^2 = \left\|(1 - \sqrt{\mu\eta})(y_k - z_k) + \sqrt{\frac{\eta}{\mu}}(v_k + \xi_k)\right\|^2$$

$$= (1 - \sqrt{\mu\eta})^2\|y_k - z_k\|^2 + 2\sqrt{\frac{\eta}{\mu}}(1 - \sqrt{\mu\eta})(y_k - z_k, v_k + \xi_k) + \frac{\eta}{\mu}\|v_k + \xi_k\|^2.$$

From above equation and (8) with $x = y_k$, we get

$$\Phi_{k+1}(y_k) = \Phi_{k+1}^* + \frac{\mu}{2}\left\{(1 - \sqrt{\mu\eta})^2\|y_k - z_k\|^2 + 2\sqrt{\frac{\eta}{\mu}}(1 - \sqrt{\mu\eta})(y_k - z_k, v_k + \xi_k)\right.$$

$$\left. + \frac{\eta}{\mu}\|v_k + \xi_k\|^2\right\}.$$

On the other hand, from the definition of the estimate sequence and (8),

$$\Phi_{k+1}(y_k) = (1 - \sqrt{\mu\eta})\left(\Phi_k^* + \frac{\mu}{2}\|y_k - z_k\|^2\right) + \sqrt{\mu\eta}(g_{I_k}(y_k) + h(x_{k+1}) + (\xi_k, y_k - x_{k+1})).$$

Therefore, from these two equations, we have

$$\Phi_{k+1}^* = (1 - \sqrt{\mu\eta})\Phi_k^* + \frac{\mu}{2}(1 - \sqrt{\mu\eta})\sqrt{\mu\eta}\|y_k - z_k\|^2 + \sqrt{\mu\eta}(g_{I_k}(y_k) + h(x_{k+1})$$

$$+ (\xi_k, y_k - x_{k+1})) - (1 - \sqrt{\mu\eta})\sqrt{\mu\eta}(y_k - z_k, v_k + \xi_k) - \frac{\eta}{2}\|v_k + \xi_k\|^2. \tag{16}$$

Since $g$ is Lipschitz smooth, we bound $f(x_{k+1})$ as follows:

$$f(x_{k+1}) \leq g(y_k) + (\nabla g(y_k), x_{k+1} - y_k) + \tfrac{L}{2}\|x_{k+1} - y_k\|^2 + h(x_{k+1}). \qquad (17)$$

Using (16), (17), (12), and $x_{k+1} - y_k = -\eta(v_k + \xi_k)$ we have

$$\mathbb{E}_{I_k}\left[f(x_{k+1}) - \Phi_{k+1}^*\right] \qquad (18)$$

$$\underset{(16),(17)}{\leq} \mathbb{E}_{I_k}\Big[(1 - \sqrt{\mu\eta})(-\Phi_k^* + g(y_k) + h(x_{k+1})) + (\nabla g(y_k), x_{k+1} - y_k)$$

$$+ \sqrt{\mu\eta}(\xi_k, x_{k+1} - y_k) + \frac{L}{2}\|x_{k+1} - y_k\|^2 - \frac{\mu}{2}(1 - \sqrt{\mu\eta})\sqrt{\mu\eta}\|y_k - z_k\|^2$$

$$+ (1 - \sqrt{\mu\eta})\sqrt{\mu\eta}(y_k - z_k, v_k + \xi_k) + \frac{\eta}{2}\|v_k + \xi_k\|^2\Big]$$

$$\underset{(12)}{=} \mathbb{E}_{I_k}\Big[(1 - \sqrt{\mu\eta})(-\Phi_k^* + g(y_k) + h(x_{k+1}) + (x_k - y_k, v_k + \xi_k)) - \eta(\nabla g(y_k), v_k + \xi_k)$$

$$- \eta\sqrt{\mu\eta}(\xi_k, v_k + \xi_k) - \frac{\mu}{2}\frac{1 - \sqrt{\mu\eta}}{\sqrt{\mu\eta}}\|y_k - x_k\|^2 + \frac{\eta}{2}(L\eta + 1)\|v_k + \xi_k\|^2\Big], \qquad (19)$$

where for the first inequality we used $\mathbb{E}_{I_k}[g_{I_k}(y_k)] = g(y_k)$. Here, we give the following

$$\mathbb{E}_{I_k}\left[g(y_k) + h(x_{k+1}) + (x_k - y_k, v_k + \xi_k)\right]$$

$$= \mathbb{E}_{I_k}\left[g(y_k) + (v_k, x_k - y_k) + h(x_{k+1}) + (\xi_k, x_k - x_{k+1}) + (\xi_k, x_{k+1} - y_k)\right]$$

$$\leq \mathbb{E}_{I_k}\left[g(x_k) - \frac{\mu}{2}\|x_k - y_k\|^2 + h(x_k) - \eta(\xi_k, v_k + \xi_k)\right], \qquad (20)$$

where for the first inequality we used $\mathbb{E}_{I_k}[v_k] = \nabla g(y_k)$ and convexity of $g$ and $h$. Thus we have

$$\mathbb{E}_{I_k}\left[f(x_{k+1}) - \Phi_{k+1}^*\right]$$

$$\underset{(19),(20)}{\leq} \mathbb{E}_{I_k}\Big[(1 - \sqrt{\mu\eta})(f(x_k) - \Phi_k^*) - \frac{\mu}{2}\frac{1 - \mu\eta}{\sqrt{\mu\eta}}\|x_k - y_k\|^2$$

$$- \eta(\nabla g(y_k) + \xi_k, v_k + \xi_k) + \frac{\eta}{2}(1 + L\eta)\|v_k + \xi_k\|^2\Big]$$

$$\underset{(13)}{\leq} \mathbb{E}_{I_k}\Big[(1 - \sqrt{\mu\eta})(f(x_k) - \Phi_k^*) - \frac{\mu}{2}\frac{1 - \mu\eta}{\sqrt{\mu\eta}}\|x_k - y_k\|^2$$

$$- \frac{\eta}{2}\|\nabla g(y_k) + \xi_k\|^2 + \frac{L\eta^2}{2}\|v_k + \xi_k\|^2 + \frac{\eta}{2}\|v_k - \nabla g(y_k)\|^2\Big]$$

$$\underset{(14),\eta\leq\frac{1}{2L}}{\leq} \mathbb{E}_{I_k}\Big[(1 - \sqrt{\mu\eta})(f(x_k) - \Phi_k^*) - \frac{\mu}{2}\frac{1 - \mu\eta}{\sqrt{\mu\eta}}\|x_k - y_k\|^2 + \eta\|v_k - \nabla g(y_k)\|^2\Big].$$

By taking expectation with respect to the history of random variables $I_1, \ldots, I_{k-1}$, the induction hypothesis finishes the proof of (10). $\qquad \square$

Our bound on the variance of $v_k$ is given in the following lemma, whose proof is in the Supplementary Material.

**Lemma 4.** *Suppose Assumption 1 holds, and let $x_* = \underset{x\in\mathbb{R}^d}{\arg\min} f(x)$. Conditioned on $y_k$, we have that*

$$\mathbb{E}_{I_k}\|v_k - \nabla g(y_k)\|^2 \leq \frac{1}{b}\frac{n - b}{n - 1}\left(2L^2\|y_k - x_k\|^2 + 8L(f(x_k) - f(x_*) + f(\tilde{x}) - f(x_*))\right). \qquad (21)$$

From (10), (21), and (9) with $x = x_*$, it follows that

$$\mathbb{E}\left[f(x_k) - f(x_*)\right] \leq (1 - \sqrt{\mu\eta})^{k-1}(\Phi_1 - f)(x_*) + \mathbb{E}\Big[\sum_{l=1}^{k-1}(1 - \sqrt{\mu\eta})^{k-1-l}$$

$$\cdot\Big\{\Big(-\frac{\mu}{2}\frac{1 - \mu\eta}{\sqrt{\mu\eta}} + \frac{n-b}{n-1}\frac{2L^2\eta}{b}\Big)\|x_l - y_l\|^2 + \frac{n-b}{n-1}\frac{8L\eta}{b}(f(x_l) - f(x_*) + f(\tilde{x}) - f(x_*))\Big\}\Big].$$

If $\eta \leq \min\left\{\frac{(pb)^2}{64}\left(\frac{n-1}{n-b}\right)^2\frac{\mu}{L^2}, \frac{1}{2L}\right\}$, then the coefficients of $\|x_l - y_l\|^2$ are non-positive for $p \leq 2$. Indeed, using

$$\eta \leq \frac{(pb)^2}{64}\left(\frac{n-1}{n-b}\right)^2\frac{\mu}{L^2} \Rightarrow \frac{n-b}{n-1}\frac{L\eta}{b} \leq \frac{p}{8}\sqrt{\mu\eta}, \quad for\ p > 0, \tag{22}$$

we get

$$-\frac{\mu}{2}\frac{1-\mu\eta}{\sqrt{\mu\eta}} + \frac{n-b}{n-1}\frac{2L^2\eta}{b} \leq -\frac{\mu}{2}\frac{1-\mu\eta}{\sqrt{\mu\eta}} + \frac{L}{2}\sqrt{\mu\eta}$$
$$= \frac{1}{2\sqrt{\mu\eta}}\left(-\mu + \mu^2\eta + \mu L\eta\right) \underset{\mu \leq L}{\leq} \frac{1}{2\sqrt{\mu\eta}}\left(-\mu + 2\mu L\eta\right) \underset{\eta \leq \frac{1}{2L}}{\leq} 0.$$

Thus, using (22) again with $p \leq 1$, we have

$$\mathbb{E}\left[f(x_k) - f(x_*)\right] \leq (1 - \sqrt{\mu\eta})^{k-1}(\Phi_1 - f)(x_*)$$
$$+ \mathbb{E}\left[\sum_{l=1}^{k-1}(1 - \sqrt{\mu\eta})^{k-1-l}p\sqrt{\mu\eta}(f(x_l) - f(x_*) + f(\tilde{x}) - f(x_*))\right]$$
$$\leq (1 - \sqrt{\mu\eta})^{k-1}(\Phi_1 - f)(x_*) + p(f(\tilde{x}) - f(x_*))$$
$$+ \mathbb{E}\left[\sum_{l=1}^{k-1}(1 - \sqrt{\mu\eta})^{k-1-l}p\sqrt{\mu\eta}(f(x_l) - f(x_*))\right], \tag{23}$$

where for the last inequality we used $\sum_{l=1}^{k-1}(1 - \sqrt{\mu\eta})^{k-1-l} \leq \sum_{t=0}^{\infty}(1 - \sqrt{\mu\eta})^t = \frac{1}{\sqrt{\mu\eta}}$.

**Theorem 1.** *Suppose Assumption 1, 2, and 3. Let* $\eta \leq \min\left\{\frac{(pb)^2}{64}\left(\frac{n-1}{n-b}\right)^2\frac{\mu}{L^2}, \frac{1}{2L}\right\}$ *and* $0 < p < 1$. *Then we have*

$$\mathbb{E}\left[f(\tilde{x}_{s+1}) - f(x_*)\right] \leq \left((1 - (1-p)\sqrt{\mu\eta})^m + \frac{p}{1-p}\right)(2+p)(f(\tilde{x}_s) - f(x_*)). \tag{24}$$

*Moreover, if* $m \geq \frac{1}{(1-p)\sqrt{\mu\eta}}\log\frac{1-p}{p}$, *then it follows that*

$$\mathbb{E}\left[f(\tilde{x}_{s+1}) - f(x_*)\right] \leq \frac{2p(2+p)}{1-p}(f(\tilde{x}_s) - f(x_*)). \tag{25}$$

From Theorem 1, we can see that for small $0 < p$, the overall complexity of Acc-Prox-SVRG (total number of component gradient evaluations to find an $\epsilon$-accurate solution) is

$$O\left(\left(n + \frac{b}{\sqrt{\mu\eta}}\right)\log\frac{1}{\epsilon}\right).$$

Thus, we have the following corollary:

**Corollary 1.** *Suppose Assumption 1, 2, and 3. Let* $p$ *be sufficiently small, as stated above, and* $\eta = \min\left\{\frac{(pb)^2}{64}\left(\frac{n-1}{n-b}\right)^2\frac{\mu}{L^2}, \frac{1}{2L}\right\}$. *If mini-batch size* $b$ *is smaller than* $\left\lceil\frac{8\sqrt{\kappa}n}{\sqrt{2}p(n-1)+8\sqrt{\kappa}}\right\rceil$, *then the learning rate* $\eta$ *is equal to* $\frac{(pb)^2}{64}\left(\frac{n-1}{n-b}\right)^2\frac{\mu}{L^2}$ *and the overall complexity is*

$$O\left(\left(n + \frac{n-b}{n-1}\kappa\right)\log\frac{1}{\epsilon}\right). \tag{26}$$

*Otherwise,* $\eta = \frac{1}{2L}$ *and the complexity becomes*

$$O\left(\left(n + b\sqrt{\kappa}\right)\log\frac{1}{\epsilon}\right). \tag{27}$$

Table 1: Comparison of overall complexity. $b_0 = \frac{8\sqrt{\kappa}n}{\sqrt{2}p(n-1)+8\sqrt{\kappa}}$.

| ProxSVRG | **AccProxSVRG** $b < \lceil b_0 \rceil$ | APG [8] | **AccProxSVRG** $b \geq \lceil b_0 \rceil$ |
|---|---|---|---|
| $O\left((n+\kappa)\log\frac{1}{\epsilon}\right)$ | $O\left(\left(n+\frac{n-b}{n-1}\kappa\right)\log\frac{1}{\epsilon}\right)$ | $O\left((n\sqrt{\kappa})\log\frac{1}{\epsilon}\right)$ | $O\left((n+b\sqrt{\kappa})\log\frac{1}{\epsilon}\right)$ |

Table 1 lists the overall complexities of the algorithms that achieve linear convergence. As seen from Table 1, the complexity of Acc-Prox-SVRG monotonically decreases with respect to $b < \lceil b_0 \rceil$, where $b_0 = \frac{8\sqrt{\kappa}n}{\sqrt{2}p(n-1)+8\sqrt{\kappa}}$ and monotonically increases when $b \geq \lceil b_0 \rceil$. Moreover, if $b = 1$, then Acc-Prox-SVRG has the same complexity as that of Prox-SVRG, while if $b = n$ then the complexity of this method is equal to that of APG. Therefore, with an appropriate mini-batch size, Acc-Prox-SVRG may outperform both Prox-SVRG and APG; even if the mini-batch is not appropriate, then Acc-Prox-SVRG is still comparable to Prox-SVRG or APG. The following overall complexity is the best possible rate of Acc-Prox-SVRG,

$$O\left(\left(n + \frac{n\kappa}{n+\sqrt{\kappa}}\right)\log\left(\frac{1}{\epsilon}\right)\right).$$

Now we give the proof of Theorem 1.

*Proof of Theorem 1.* We denote $E[f(x_k) - f(x_*)]$ by $V_k$, and we use $W_k$ to denote the last expression in (23). Thus, for $k \geq 1$, $V_k \leq W_k$. For $k \geq 2$, we have

$$W_k = (1 - \sqrt{\mu\eta})\left\{(1 - \sqrt{\mu\eta})^{k-2}(\Phi_1 - f)(x_*) + pV_1 + \sum_{l=1}^{k-2}(1 - \sqrt{\mu\eta})^{k-2-l}p\sqrt{\mu\eta}\, V_l\right\}$$
$$+ p\sqrt{\mu\eta}\, V_{k-1} + p\sqrt{\mu\eta}\, V_1 \leq (1 - \sqrt{\mu\eta}(1-p))W_{k-1} + p\sqrt{\mu\eta}\, W_1.$$

Since $0 < \sqrt{\mu\eta}(1-p) < 1$, the above inequality leads to

$$W_k = \left((1 - (1-p)\sqrt{\mu\eta})^{k-1} + \frac{p}{1-p}\right)W_1. \tag{28}$$

From the strong convexity of $g$ (and $f$), we can see

$$W_1 = (1+p)(f(\tilde{x}) - f(x_*)) + \frac{\mu}{2}\|\tilde{x} - x_*\|^2 \leq (2+p)(f(\tilde{x}) - f(x_*)).$$

Thus, for $k \geq 2$, we have

$$V_k \leq W_k \leq \left((1 - (1-p)\sqrt{\mu\eta})^{k-1} + \frac{p}{1-p}\right)(2+p)(f(\tilde{x}) - f(x_*)),$$

and that is exactly (24). Using $\log(1-\alpha) \leq -\alpha$ and $m \geq \frac{1}{(1-p)\sqrt{\mu\eta}}\log\frac{1-p}{p}$, we have

$$\log(1 - (1-p)\sqrt{\mu\eta})^m \leq -m(1-p)\sqrt{\mu\eta} \leq -\log\frac{1-p}{p},$$

so that

$$(1 - (1-p)\sqrt{\mu\eta})^m \leq \frac{p}{1-p}.$$

This finishes the proof of Theorem 1. $\qquad\square$

## 4  Numerical Experiments

In this section, we compare Acc-Prox-SVRG with Prox-SVRG and APG on $L_1$-regularized multi-class logistic regression with the regularization parameter $\lambda$. Table 2 provides details of the datasets

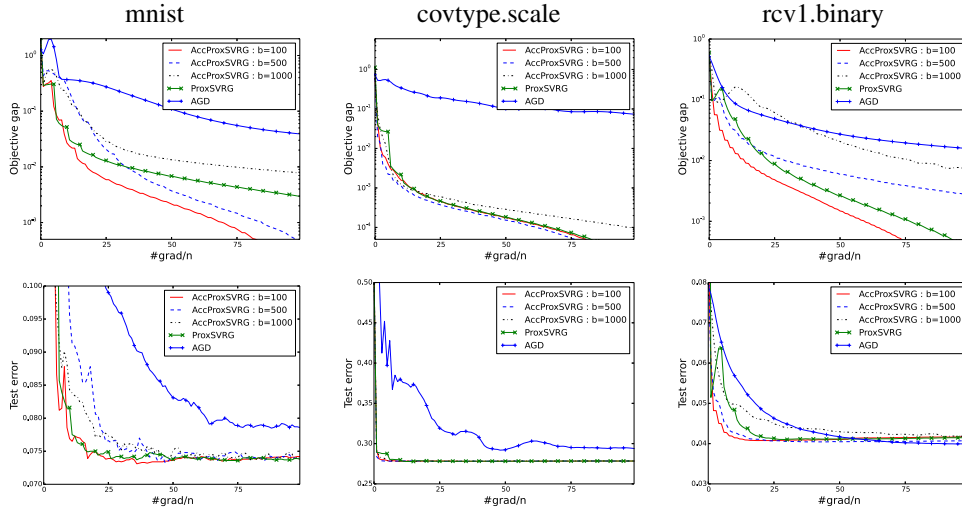

Figure 2: Comparison of Acc-Prox-SVRG with Prox-SVRG and APG. Top: Objective gap of $L_1$ regularized multi-class logistic regression. Bottom: Test error rates.

and regularization parameters utilized in our experiments. These datasets can be found at the LIBSVM website[1]. The best choice of mini-batch size is $b = \lceil b_0 \rceil$, which allows us to take a large learning rate, $\eta = \frac{1}{2L}$. Therefore, we have $m \geq O(\sqrt{\kappa})$ and $\beta_k = \frac{\sqrt{2\kappa}-1}{\sqrt{2\kappa}+1}$. When the number of components $n$ is very large compared with $\sqrt{\kappa}$, we see that $b_0 = O(\sqrt{\kappa})$; for this, we set $m = \delta b$ ($\delta \in \{0.1, 1.0, 10\}$) and $\beta_k = \frac{b-2}{b+2}$ varying $b$ in the set $\{100, 500, 1000\}$. We ran AccProx-SVRG using values of $\eta$ from the range $\{0.01, 0.05, 0.1, 0.5, 1.0, 5.0, 10.0\}$, and we chose the best $\eta$ in each mini-batch setting.

Figure 2 compares Acc-Prox-SVRG with Prox-SVRG and APG. The horizontal axis is the number of single-component gradient evaluations. For Acc-Prox-SVRG, each iteration computes the $2b$ gradients, and at the beginning of each stage, the $n$ component gradients are evaluated. For ProxSVRG, each iteration computes the two gradients, and at the beginning of each stage, the $n$ gradients are evaluated. For APG, each iteration evaluates $n$ gradients.

Table 2: Details of data sets and regularization parameters.

| Dataset | classes | Training size | Testing size | Features | $\lambda$ |
|---|---|---|---|---|---|
| mnist | 10 | 60,000 | 10,000 | 780 | $10^{-5}$ |
| covtype.scale | 7 | 522,910 | 58,102 | 54 | $10^{-6}$ |
| rcv1.binary | 2 | 20,242 | 677,399 | 47,236 | $10^{-5}$ |

As can be seen from Figure 2, Acc-Prox-SVRG with good values of $b$ performs better than or is comparable to Prox-SVRG and is much better than results for APG. On the other hand, for relatively large $b$, Acc-Prox-SVRG may perform worse because of an overestimation of $b_0$, and hence the worse estimates of $m$ and $\beta_k$.

## 5  Conclusion

We have introduced a method incorporating Nesterov's acceleration method and a variance reduction technique of SVRG in the mini-batch setting. We prove that the overall complexity of our method, with an appropriate mini-batch size, is more efficient than both Prox-SVRG and APG; even when mini-batch size is not appropriate, our method is still comparable to APG or Prox-SVRG. In addition, the gradient evaluations for each mini-batch can be parallelized [3, 10, 11] when using our method; hence, it performs much faster in a distributed framework.

## Footnotes

[1]http://www.csie.ntu.edu.tw/ cjlin/libsvmtools/datasets/

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
