[Supplementary Material]

# Supplementary Materials: Stochastic Proximal Gradient Descent with Acceleration Techniques

**Atsushi Nitanda**
NTT DATA Mathematical Systems Inc.
1F Shinanomachi Rengakan, 35,
Shinanomachi, Shinjuku-ku, Tokyo,
160-0016, Japan
nitanda@msi.co.jp

## 1 Proofs of the first half of Lemma 1 and Lemma 2

Before we prove lemmas, we again give the definition of estimate sequence and its basic properties.

$$\Phi_1(x) = f(x_1) + \frac{\mu}{2}\|x - x_1\|^2,$$

$$\Phi_{k+1}(x) = (1 - \sqrt{\mu\eta})\Phi_k(x) + \sqrt{\mu\eta}(g_{I_k}(y_k) + (v_k, x - y_k) + \frac{\mu}{2}\|x - y_k\|^2$$
$$+ h(x_{k+1}) + (\xi_k, x - x_{k+1})), \quad k \geq 1.$$

We set

$$\Phi_k^* = \min_{x \in \mathbb{R}^d} \Phi_k(x), \quad z_k = \arg\min_{x \in \mathbb{R}^d} \Phi_k(x).$$

Since $\nabla^2\Phi_k(x) = \mu I_n$, it follows that for $\forall x \in \mathbb{R}^d$

$$\Phi_k(x) = \frac{\mu}{2}\|x - z_k\|^2 + \Phi_k^*. \tag{1}$$

Now we give the proof of first half of Lemma 1.

*Proof of the first half of Lemma 1.* We see that (9) of Lemma 1 is true for $k = 1$, and we assume it is true for $k$. From the definition of estimate sequence, we have

$$\mathbb{E}[\Phi_{k+1}(x)] = (1 - \sqrt{\mu\eta})\mathbb{E}[\Phi_k(x)] + \sqrt{\mu\eta}\,\mathbb{E}[g_{I_k}(y_k) + (v_k, x - y_k)$$
$$+ \frac{\mu}{2}\|x - y_k\|^2 + h(x_{k+1}) + (\xi_k, x - x_{k+1})]$$
$$\leq (1 - \sqrt{\mu\eta})f(x) + (1 - \sqrt{\mu\eta})^k(\Phi_1 - f)(x)$$
$$+ \sqrt{\mu\eta}\,\mathbb{E}\left[g(y_k) + (\nabla g(y_k), x - y_k) + \frac{\mu}{2}\|x - y_k\|^2 + h(x_{k+1}) + (\xi_k, x - x_{k+1})\right]$$
$$\leq (1 - \sqrt{\mu\eta})f(x) + (1 - \sqrt{\mu\eta})^k(\Phi_1 - f)(x) + \sqrt{\mu\eta}(g(x) + h(x))$$
$$= f(x) + (1 - \sqrt{\mu\eta})^k(\Phi_1 - f)(x),$$

where for the first inequality we used induction hypothesis, $\mathbb{E}_{I_k}[g_{I_k}(y_k)] = \mathbb{E}[g(y_k)]$ and $\mathbb{E}_{I_k}[v_k] = \mathbb{E}[\nabla g(y_k)]$, for the last inequality we used the convexity of $g$ and $h$. Hence, the first half of Lemma (9) follows. $\qquad\square$

Next, we give the proof of Lemma 2.

*Proof of Lemma 2.* From the definition of estimate sequence and (1), we have that for $k \geq 1$

$$\frac{\mu}{2}\|x - z_{k+1}\|^2 + \Phi_{k+1}^* = (1 - \sqrt{\mu\eta})\left(\frac{\mu}{2}\|x - z_k\|^2 + \Phi_k^*\right) + \sqrt{\mu\eta}\Big(g_{I_k}(y_k) + (v_k, x - y_k)$$
$$+ \frac{\mu}{2}\|x - y_k\|^2 + h(x_{k+1}) + (\xi_k, x - x_{k+1})\Big).$$

By differentiating at $y_k$ this equality, we obtain

$$\mu(y_k - z_{k+1}) = (1 - \sqrt{\mu\eta})\mu(y_k - z_k) + \sqrt{\mu\eta}(v_k + \xi_k).$$

Hence, we have

$$z_{k+1} = (1 - \sqrt{\mu\eta})z_k + \sqrt{\mu\eta}y_k - \sqrt{\frac{\eta}{\mu}}(v_k + \xi_k),$$

and that is exactly (11) of Lemma 2. Next, we prove (12) of Lemma 2 by induction. It is true for $k = 1$. We assume it is true for $k$, then it follows from (11) of Lemma 2 that,

$$z_{k+1} - y_{k+1} = (1 - \sqrt{\mu\eta})z_k + \sqrt{\mu\eta}y_k - \sqrt{\frac{\eta}{\mu}}(v_k + \xi_k) - y_{k+1}$$
$$= \frac{1}{\sqrt{\mu\eta}}(y_k - \eta(v_k + \xi_k)) - \frac{1 - \sqrt{\mu\eta}}{\sqrt{\mu\eta}}x_k - y_{k+1}$$
$$= \frac{1}{\sqrt{\mu\eta}}x_{k+1} - \frac{1 - \sqrt{\mu\eta}}{\sqrt{\mu\eta}}x_k - y_{k+1}.$$

From the update rule of $y_{k+1}$, we have

$$-\frac{1 - \sqrt{\mu\eta}}{\sqrt{\mu\eta}}x_k = \frac{1 + \sqrt{\mu\eta}}{\sqrt{\mu\eta}}\left(y_{k+1} - \left(1 + \frac{1 - \sqrt{\mu\eta}}{1 + \sqrt{\mu\eta}}\right)x_{k+1}\right)$$
$$= \frac{1 + \sqrt{\mu\eta}}{\sqrt{\mu\eta}}y_{k+1} - \frac{2}{\sqrt{\mu\eta}}x_{k+1}.$$

Hence, we get

$$z_{k+1} - y_{k+1} = \frac{1}{\sqrt{\mu\eta}}(y_{k+1} - x_{k+1}).$$

$\square$

## 2 Proof of Lemma 3

We prove Lemma 3.

*Proof of Lemma 3.* Averaging

$$(\nabla g(y_k) + \xi_k, v_k + \xi_k) = \|\nabla g(y_k) + \xi_k\|^2 + (\nabla g(y_k) + \xi_k, v_k - \nabla g(y_k))$$

and

$$(\nabla g(y_k) + \xi_k, v_k + \xi_k) = \|v_k + \xi_k\|^2 + (v_k + \xi_k, \nabla g(y_k) - v_k),$$

we get (13) of Lemma 3:

$$(\nabla g(y_k) + \xi_k, v_k + \xi_k) = \frac{1}{2}\left(\|\nabla g(y_k) + \xi_k\|^2 + \|v_k + \xi_k\|^2 - \|\nabla g(y_k) - v_k\|^2\right).$$

(14) of Lemma 3 is shown as follows:

$$\|v_k + \xi_k\|^2 = \|v_k + \xi_k + \nabla g(y_k) + \xi_k - (\nabla g(y_k) + \xi_k)\|^2$$
$$= \|\nabla g(y_k) + \xi_k\|^2 + 2(\nabla g(y_k) + \xi_k, v_k - \nabla g(y_k)) + \|v_k - \nabla g(y_k)\|^2$$
$$\leq 2(\|\nabla g(y_k) + \xi_k\|^2 + \|v_k - \nabla g(y_k)\|^2).$$

In the last inequality, we use

$$|(a,b)| \leq \frac{\|a\|^2 + \|b\|^2}{2}, \quad for\ \forall a, \forall b \in \mathbb{R}^d.$$

Inequality (15) of Lemma 3 can be proved in a similar way. $\square$

## 3  Proof of Lemma 4

Lemma 4 is the key lemma which give a bound on the variance. Now we give the proof.

*Proof of Lemma 4.* We set $v_j^1 = \nabla g_j(y_k) - \nabla g_j(\tilde{x}) + \tilde{v}$. Since

$$v_k = \frac{1}{b} \sum_{j \in I_k} v_j^1,$$

conditional variance of $v_k$ is as follows (see [1, p.183])

$$\mathbb{E}_{I_k} \|v_k - \nabla g(y_k)\|^2 = \frac{1}{b} \frac{n-b}{n-1} \mathbb{E}_j \|v_j^1 - \nabla g(y_k)\|^2,$$

where expectation in right hand side is taken with respect to $j \in \{1, \ldots, n\}$. Therefore, it suffices to prove that

$$\mathbb{E}_j \|v_j^1 - \nabla g(y_k)\|^2 \leq 2L^2 \|y_k - x_k\|^2 + 8L(f(x_k) - f(x_*) + f(\tilde{x}) - f(x_*)). \tag{2}$$

For $i \in \{1, \ldots, n\}$, we set

$$\phi_i(x) = g_i(x) - (g_i(x_*) + (\nabla g_i(x_*), x - x_*)).$$

We have that $\phi_i(x_*) = \min_x \phi_i(x)$ since $\nabla \phi_i(x_*) = 0$ and convexity of $\phi_i$. Since $\nabla \phi_i$ is Lipschitz continuous with $L$, it follows that (see [2, Theorem 2.1.5])

$$\frac{1}{2L} \|\nabla \phi_i(x)\|^2 \leq \phi_i(x) - \phi_i(x_*) = \phi_i(x),$$

Thus,

$$\|\nabla g_i(x) - \nabla g_i(x_*)\|^2 \leq 2L(g_i(x) - g_i(x_*) - (\nabla g_i(x_*), x - x_*)).$$

Averaging from $i = 1$ to $n$, we have

$$\frac{1}{n} \sum_{i=1}^{n} \|\nabla g_i(x) - \nabla g_i(x_*)\|^2 \leq 2L(g(x) - g(x_*) - (\nabla g(x_*), x - x_*)).$$

By the optimality of $x_*$, $-\nabla g(x_*)$ is a subgradient of $h$ at $x_*$, so that

$$(-\nabla g(x_*), x - x_*) \leq h(x) - h(x_*).$$

Hence we get

$$\frac{1}{n} \sum_{i=1}^{n} \|\nabla g_i(x) - \nabla g_i(x_*)\|^2 \leq 2L(g(x) - g(x_*) + h(x) - h(x_*)) = 2L(f(x) - f(x_*)). \tag{3}$$

We now bound left hand side of (2) as follows:

$$
\begin{aligned}
\mathbb{E}_j &\|v_j^1 - \nabla g(y_k)\|^2 \\
&= \mathbb{E}_j \|\nabla g_j(y_k) - \nabla g_j(\tilde{x}) - (\nabla g(y_k) - \nabla g(\tilde{x}))\|^2 \\
&\leq \mathbb{E}_j \|\nabla g_j(y_k) - \nabla g_j(\tilde{x})\|^2 \\
&\leq 2\mathbb{E}_j \|\nabla g_j(y_k) - \nabla g_j(x_k)\|^2 + 4\mathbb{E}_j \|\nabla g_j(x_k) - \nabla g_j(x_*)\|^2 + 4\mathbb{E}_j \|\nabla g_j(x_*) - \nabla g_j(\tilde{x})\|^2 \\
&\leq 2L^2 \|y_k - x_k\|^2 + 8L(f(x_k) - f(x_*) + f(\tilde{x}) - f(x_*)),
\end{aligned}
$$

where for the first inequality we used $\mathbb{E}\|\zeta - \mathbb{E}\zeta\|^2 \leq \mathbb{E}\|\zeta\|^2$ for any random vector $\zeta$, for the second inequality, we used $\|a + b\|^2 \leq 2\|a\|^2 + 2\|b\|^2$, and for the last inequality, we used $L$-Lipschitz continuity and (3). This finishes the proof of lemma. $\square$