[Reviews · NeurIPS 2014]

Submitted by Assigned_Reviewer_6

This paper seems to essentially combine three published ideas:
accelerated gradient, the stochastic gradient variance reduction
technique of Johnson and Zhang, and variance reduction via minibatching.
Hence, on a conceptual level at least, it's a fairly incremental paper (I
don't want to minimize the effort that may have gone into developing the
convergence proof). With this said, it's well-done, mostly well-written,
and has good theoretical and experimental results.

In terms of quality, originality and significance, it's as I said above:
they're combining pre-existing ideas, but doing it well, and included a
convergence proof with a slightly improved rate over "the competition".
The writing style is clear, although I might argue with the authors'
decision to include large blocks of proofs on pages 4-6. It doesn't look
good aesthetically, and most readers will be more interested in the
theorem statements then the proofs. Hence, the the inclusion of the
proofs in-line, instead of only in the supplementary material, breaks up
the flow of the paper. This is a minor quibble, however.

The experiments are solid, although the performance improvement over
ProxSVRG looks rather small. I'm glad that they include bad choices of
the minibatch size in the RCV1 experiments, and also that both objective
values, and test error rates, are plotted. It would be nice if additional
baseline algorithms could be included to explore the questions of how
much of the performance improvement of their proposed algorithm is due to
AG vs PGD, how much to the Johnson and Zhang-style variance reduction,
and how much to the inclusion of minibatching. This isn't a criticism,
just a suggestion--the experiments look good.
Summary: An somewhat incremental, but quality submission, which combines existing
ideas to yield an algorithm with slightly better bounds than those
previously published, with experiments verifying that it achieves at
least modest gains in practice.

Submitted by Assigned_Reviewer_36

This paper proposes an acceleration technique for Stochastic Proximal Gradient Descent (SPGD). The acceleration is a combination of mini-batch optimization and Nesterov accelerated gradient descent. The authors show that if the batch size could be appropriately chosen, then the convergence rate of the proposed algorithm will be faster than either Prox-SVRG or APG. The technical contribution of this paper, as far as I am concerned, is solid. The experimental result shows that the proposed algorithm is faster than both ProxSVRG and APG.

The proofs given in Section 3 are difficult to follow. The authors should add more comments and remarks to explain the meaning of defined notations, e.g. p, z_k, \Phi and \Phi^*. When b is chosen to be b_0, the overall complexity of the algorithm is on the order of

n + n*\kappa/(n+\sqrt{\kappa}) ...(*)

This rate should be explicitly given at the end of Section 3, as this is the best possible rate that the proposed algorithm could achieve, assuming a perfect knowledge of the optimal batch size. It is worth mentioning that Shalev-Shwartz and Zhang have proposed another accelerated version of the SDCA algorithm [1], whose convergence rate is

n + \sqrt{n*\kappa} ...(**)

which is at least as fast as (*). Lin et al. have also proposed an algorithm [2] achieving the same rate of (**). The authors may want to compare their algorithm with [1] and [2], either theoretically or experimentally.

Overall, I think this is an interesting paper. The technical part is solid, though the improvement is slightly incremental.

[1] Shai Shalev-Shwartz and Tong Zhang. Accelerated Proximal Stochastic Dual Coordinate Ascent for Regularized Loss Minimization

[2] Qihang Lin, Zhaosong Lu, and Lin Xiao, An Accelerated Proximal Coordinate Gradient Method and its Application to Regularized Empirical Risk Minimization
Summary: Overall, I think this is an interesting paper. The technical part is solid, though the improvement is slightly incremental.

Submitted by Assigned_Reviewer_38

The paper presents an accelerated mini-batch randomized incremental gradient descent algorithm for composite minimization. The composite objective function is supposed to decompose into the sum of differentiable, Lipschitz-continuous terms and a proximal term. The contributions can be summarized as a two-fold extension of stochastic proximal variance reduction gradient descent (Prox-SVRG; Johnson&Zhang, Xiao and Zhang), using both an accelerated-gradient-type acceleration (Nesterov's) and a mini-batch setting.

The obtained complexity estimate is in between O((n+ alpha*kappa)log(1/epsilon)) and O((n+gamma*sqrt(kappa))log(1/epsilon)) depending on the appropriate choice of the mini-batch size wrt to the "optimal" (knowing the true condition number) mini-batch size (n is the number of terms in the diff.-part of the objective, kappa the condition number, and epsilon the accuracy). At worst, the proposed method performs as well as the accelerated proximal gradient decent and the Prox-SVRG. With appropriate mini-batch size, it potentially outperform both.

Illustrative preliminary experiments are presented for learning with a multinomial logistic loss and an L1 regularization penalty. These experiments suggest that the proposed algorithm can outperform the competing methods with appropriate choice of the mini-batch size.

Detailed comments:

- the term "stochastic" is misleading here, since the composite optimization problem in Eq. 1 is completely deterministic and does not involve any expectation. Algorithms proceeding by randomly picking subset of terms in a large sum appearing the optimization objective are usually referred to "randomized gradient descent" algorithms. See [Bertsekas, "Nonlinear Programming", 2003] for reference in terminology. This stands in contrast to "stochastic gradient descent" algorithms whose purpose it to minimize an objective involving an expectation.

- last of paragraph of page 2 is unclear, probably because of unfortunate copy-paste or edits; please correct.

- throughout the paper, expectations are used, sometimes without conditioning to any filtration, sometimes with conditioning to a particular filtration, but always with the same notation. While the derivations seem to be correct, this is very confusing. Please use a less ambiguous notation, for instance E_t for E ( . | F_t).

- Section 3 could start with a proof outline. Once the 4 lemmas are proved, the mechanics of the proof of Theorem 1 are quite clear and easy to follow. It would probably make sense to state the lemmas without proof (with proofs deferred to Appendix), and use the space to comment and illustrate what's going on in the proof. In particular, it would be interesting to include illustrations of the interpretation of the behaviour of the estimate sequences along the iterations for the convergence of the proposed algorithm.

- Section 4 should be improved. A major issue of mini-batch approaches that limited their use in practice is the tuning of the mini-batch size. Several heuristics could be explored and compared here, since this parameter has clearly a major impact on relative performance wrt competing methods. It would also be interesting to compare the algorithms for several values of the regularization parameter. A default option could be the value that leads to the best validation error. It's quite shocking here (in a submission to a ML conference) that the value was set to a default value uniformly over all datasets, regardless of the generalization performance. Finally, it would be interesting to investigate the behaviour of the algorithms on a large dataset (n >> 1) with high-dimensional and dense features (p >>1, and dense vectors). This case is not covered by the datasets considered, but would clearly be worthwhile to consider in order to compare the algorithms.
Summary: The paper proposes an interesting new algorithm for large-scale learning with composite objectives, using randomized incremental algorithms. The proposed method fills a gap in the literature, and can potentially have practical interest wrt accelerated-gradient algorithms and Prox-SVRG algorithms. Yet, the exposition and content could be improved; see detailed comments.
Author Feedback
Author rebuttal: We thank reviewers for helpful comments and suggestions.

------------------
To reviewer_36:
------------------

- "The acceleration is a combination of mini-batch optimization and Nesterov accelerated gradient descent. "

Our method also incorporates the variance reduction technique of SVRG.

- "When b is chosen to be b_0, the overall complexity of the algorithm is on the order of
n + n*\kappa/(n+\sqrt{\kappa}) ...(*)
This rate should be explicitly given at the end of Section 3,"

We will revise our paper according to the suggestion.

------------------
To reviewer_38:
------------------

- "last of paragraph of page 2 is unclear, probably because of unfortunate copy-paste or edits; please correct."

We will fix this.

- "throughout the paper, expectations are used, sometimes without conditioning to any filtration, sometimes with conditioning to a particular filtration, but always with the same notation. While the derivations seem to be correct, this is very confusing. Please use a less ambiguous notation, for instance E_t for E ( . | F_t)."

We will revise our paper according to the suggestion, which will make the paper easier to read.

- "It would also be interesting to compare the algorithms for several values of the regularization parameter. A default option could be the value that leads to the best validation error. It's quite shocking here (in a submission to a ML conference) that the value was set to a default value uniformly over all datasets, regardless of the generalization performance."

We have experimented with several values of regularization parameters \lambda \in \{ 1e-4, 1e-5, 1e-6 \}.
In every case, our method with good values of mini-batch size performs better than or is comparable to Prox-SVRG and is much better than results for APG.
We will revise experimental results with \lambda which leads to the best generalization performance.

------------------
To reviewer_6:
------------------

Thanks for positive feedback.